# Cellular and Molecular Mechanisms Mediating Methylmercury Neurotoxicity and Neuroinflammation

**DOI:** 10.3390/ijms22063101

**Published:** 2021-03-18

**Authors:** João P. Novo, Beatriz Martins, Ramon S. Raposo, Frederico C. Pereira, Reinaldo B. Oriá, João O. Malva, Carlos Fontes-Ribeiro

**Affiliations:** 1Institute for Clinical and Biomedical Research (iCBR), Center for Innovative Biomedicine and Biotechnology (CIBB), and Institute of Pharmacology and Experimental Therapeutics, Faculty of Medicine, University of Coimbra, 3000-548 Coimbra, Portugal; jpfcnovo@gmail.com (J.P.N.); beatrizcorterealmartins@gmail.com (B.M.); rraposo@uc.pt (R.S.R.); fredcp@ci.uc.pt (F.C.P.); 2Experimental Biology Core, University of Fortaleza, Health Sciences, Fortaleza 60110-001, Brazil; 3Laboratory of Tissue Healing, Ontogeny and Nutrition, Department of Morphology and Institute of Biomedicine, School of Medicine, Federal University of Ceará, Fortaleza 60430-270, Brazil; oria@ufc.br

**Keywords:** mercury cycle, Methylmercury, neuroinflammation, neurotoxicity, microglia, oligodendrocytes, astrocytes, neurons, oxidative stress

## Abstract

Methylmercury (MeHg) toxicity is a major environmental concern. In the aquatic reservoir, MeHg bioaccumulates along the food chain until it is consumed by riverine populations. There has been much interest in the neurotoxicity of MeHg due to recent environmental disasters. Studies have also addressed the implications of long-term MeHg exposure for humans. The central nervous system is particularly susceptible to the deleterious effects of MeHg, as evidenced by clinical symptoms and histopathological changes in poisoned humans. In vitro and in vivo studies have been crucial in deciphering the molecular mechanisms underlying MeHg-induced neurotoxicity. A collection of cellular and molecular alterations including cytokine release, oxidative stress, mitochondrial dysfunction, Ca^2+^ and glutamate dyshomeostasis, and cell death mechanisms are important consequences of brain cells exposure to MeHg. The purpose of this review is to organize an overview of the mercury cycle and MeHg poisoning events and to summarize data from cellular, animal, and human studies focusing on MeHg effects in neurons and glial cells. This review proposes an up-to-date compendium that will serve as a starting point for further studies and a consultation reference of published studies.

## 1. Introduction

Mercury (Hg) is a toxic metal with recognized adverse health impacts [1,2,3]. It is ubiquitously distributed across the globe and considered to be one of the major environmental pollutants, widely used by humans for centuries in several activities such as agriculture, industry, and medicine [4,5,6,7,8]. 

The US Government Agency for Toxic Substances and Disease Registry considers Hg to be the third most toxic substance on the planet [9]. In the same line, the World Health Organization considers Hg as one of the ten chemicals of major public health concern [10]. Due to its nature, it can be reduced, but never completely removed or destroyed [4,10].

Hg may exist in a variety of forms including elemental Hg (Hg0), inorganic, and organic compounds [4,8,9,10]. Elemental Hg is liquid at room temperature and is easily converted to vapor and emitted into the atmosphere [4,9]. Inorganic Hg occurs in two forms (mercurous, Hg+; mercuric, Hg2+) that are usually in the solid state as salts. Organic mercury is produced by the combination with carbon, generating MeHg and ethylmercury [4,7]. Dimethylmercury is among the most dangerous mercury compounds. This is a highly toxic and deadly compound if in direct contact with skin or even through latex gloves [9]. Even though all forms of mercury have toxicity capacity, alkylmercury compounds are of particular concern because of their easy penetration of biological membranes, effective bioaccumulation, and long-term removal from tissues [11].

## 2. Environmental Impact of Mercury

### 2.1. Mercury Cycle

Mercury is naturally present at low levels in the environment [12]. However, anthropogenic activities may release large amounts of Hg into the environment, leading to widespread pollution. The natural causes of Hg emissions and distribution comprise volcanic and geothermal activity, land use, biomass burning, and meteorological events [6]. On the other hand, coal combustion, fluorescent lighting, cement production, amalgam fillings, crematoria, paper production, gold mining, perfume and fur industries are some examples of anthropogenic sources that add up to the natural counterparts [2,6,9,13,14,15]. Despite the fact that Hg originates from both natural and human sources, reports attribute environmental mercury contamination mainly due to human actions [8].

Hg is found throughout the biosphere, arising in water, soil, air, and living beings [6]. The major earth reservoirs of Hg include the atmosphere, the terrestrial ecosystems, and the aquatic compartments (which include both freshwater and the oceans). Hg is actively exchanged between these pools, creating the so-called Mercury Cycle [16]. Environmental Hg net pools and interactions are interchangeable and highly modulated by human activities. The Mercury Cycle experiences disturbances derived from alterations in anthropogenic activities like soil use and climate changes which mobilize Hg that has been accumulated and naturally stabilized in each environmental reservoir [16].

Due to its volatile nature, under normal conditions, Hg vaporizes and spreads through atmospheric processes [9,17,18]. Since the industrial revolution, human activities substantially increased the emissions of atmospheric gaseous mercury, almost tripling the amounts of Hg in the atmosphere [8,9,18]. It is the consensus that human influences have drastically changed the global cycle of Hg [14]. These alterations have been accelerated with rapid urban human population growth and industrialization since the 1970s [16]. Hg emanated into the atmosphere has a long lifetime (ranging between 6 and 12 months), facilitating its global dissemination [14]. In this way, Hg is ultimately deposited onto land or into water, even in remote areas [6,18]. Once deposited, Hg can be methylated, generating MeHg with high toxicity properties [4,6,19]. 

In the terrestrial reservoir, the largest Hg pool is located in soil [18], with the potential to enter the food chain via vegetables and livestock [9]. Nevertheless, vegetation is a low-level source of Hg [18]. Mangrove ecosystems, which constitute a mixture of both terrestrial and aquatic interactions, greatly contribute to Hg storage and transport [20]. Anthropogenic actions have considerably enhanced the accumulation of Hg into this ecosystem. Additionally, the terrestrial compartment is a significant indirect source of atmospheric Hg to aquatic systems via soil drainage [18]. However, unlike air emissions releases to land do not immediately circulate on a global scale [14].

The largest bulk of Hg inputs to water compartments comes from atmospheric deposition [21]. Oceans constitute a considerable fraction of the global Hg reservoir [16]. Hg may be transported by river and ocean circulation and settle in sediments or may be dissolved to its gaseous form and evade back to the atmosphere [16,21]. Curiously, reports indicate that Hg concentrations are lower in waters with more salinity [1,22]. Higher levels of Hg have been detected in freshwater environments than coastline waters and the open ocean, which is consistent with the fact that Hg releases to water do not instantly spread worldwide [14,17,20]. Similarly to the terrestrial reservoir, MeHg production in aquatic ecosystems emerges mainly from bacterial methylation of Hg [16].

### 2.2. Methylmercury Biomagnification

Most attention regarding Hg pollution is focused on MeHg [18]. MeHg is not only the most common form of organic Hg found in nature but also the main source of organic Hg in ecosystems, with an ability to accumulate in the aquatic food chain [9,18]. Since MeHg is created in the environment, its dynamics may be different from those of inorganic Hg [14]. Wetlands are seen as places for MeHg production in sediments, which can be transported to coastal waters and may end up entering the food chain [20]. In fact, this environment has ideal conditions for Hg methylation which include elevated temperature, low pH levels, high carbon availability, and substrate to support bacterial methylation activity [16,20,23]. Additionally, sunlight radiation and industrial activities are also routes for MeHg formation [5,7].

Afterwards, MeHg bioaccumulates in the marine food-web [7]. It is important to notice that water chemistry is crucial to determine MeHg concentrations in the food-chain due to the regulation of MeHg uptake at the bottom of the food chain [24]. 

Starting with phytoplankton, there is already an accumulation of MeHg above the concentration of the surrounding waters by passively diffusing across the cell membrane [23,24,25,26]. Then, MeHg accumulates from invertebrates like zooplankton and clams to mid-trophic fishes and marine mammals such as seals, reaching the highest concentrations in the top predators, a process known as biomagnification [5,8,23]. Moreover, MeHg accumulated in organisms may be also produced by the hosts themselves in a process that involves both coenzymes and microbial activities in their body or gut [27]. The conversion of Hg to MeHg has also been detected in vivo in marine species [27]. The accumulation of MeHg in muscle tissues of predatory fish and shellfish constitutes an issue for humans that consume seafood, establishing this route as the main source of exposure to humans [7,21,28]. The exacerbated amounts of MeHg in higher trophic levels of the marine food web differ greatly from other metals whose concentrations either decline or stabilize with increasing levels in the aquatic food chain [24]. 

The fact that Hg concentration in vegetation is low makes terrestrial herbivores less exposed to this toxic metal. In fact, bioaccumulation of MeHg is less pronounced in terrestrial food chains than the aquatic equivalents. Because of this, the terrestrial pathway is not a relevant supplier of Hg to animals at higher levels of the food chain. Accordingly, piscivorous predators have higher MeHg loads than solely terrestrial predators [18].

### 2.3. The Minamata Disaster

The deadly consequences of organic mercury compounds have been demonstrated by mass-poisonings of human populations [29]. A severe MeHg intoxication occurred in Minamata and neighboring communities in Japan between the 1950s and 1960s [6,28,30]. This was the first and most famous event of severe MeHg poisoning caused by anthropogenic activities, later known as Minamata Disease [7,29,31]. The pollutant was produced from mercury as a by-product of acetaldehyde and vinyl compounds manufacturing by Chisso Co. Ltd. in Minamata City and discharged into the Minamata Bay [7,28,30]. This way, the population of the Minamata area that largely depended on fish and shellfish consumption from the Minamata Bay was contaminated with MeHg [7,28,31,32]. The Minamata disaster was a stereotypical case of biomagnification of MeHg through trophic levels that ended up in human exposure [29].

Affected individuals exhibited symptoms that indicate neurological alterations which include ataxia, visual field and hearing alterations, dysarthria, paresthesias in the distal parts of extremities, disequilibrium, gait impairment, tremors, muscle weakness, atypical eye movement, and seizures [28,29,30,31,33]. Occasionally, mental disorders and disturbances of smell and taste were also present [28]. Beyond injuries of the nervous system (the primary target of MeHg), other organs suffer minor damages, including kidneys and liver, pancreatic islet alterations, lymph node atrophy, and gastrointestinal tract inflammation [5,33]. Ultimately, MeHg led to the death of some poisoned locals [29]. 

Furthermore, MeHg is associated with fetotoxicity translating into miscarriage, stillbirth, low birth weights, and spontaneous abortions [9]. A significant number of fetuses was exposed to MeHg in utero during this period [31]. These children were born with conditions of cerebral palsy, intellectual disability, ataxia, and hypersalivation [28,31]. This is explained by the fact that mercury amounts in cord blood tend to be greater than in maternal blood since Hg is known to easily cross the placenta [9,34]. The fetal brain is more vulnerable to the noxious effects of MeHg, which translates into a disruption of the cerebral architecture and severe mental deficits, as previously mentioned [5]. Additionally, recent in vitro studies with human trophoblastic cells have been providing insights into pregnancy-related diseases caused by MeHg [35]. Another striking finding was the unexpectedly low numbers of males born in Minamata following the environmental disaster [7]. Over 50 cases of Fetal Minamata Disease were diagnosed, revealing a special sensitivity of fetal toxicity even in the absence of mother’s symptoms [7,8,28,29,31]. Fetal exposure to MeHg in Minamata has carried consequences until the present day, since older adult patients born in Minamata in the 1950s exhibit steeper declines in physical and cognitive functions than subjects of similar ages born in the same period but in non-affected areas [31].

Alongside clinical symptoms of MeHg intoxication in the population, locals reported strange phenomena namely agglomerations of fish on the surface of the Bay with outlandish swimming behaviors, sea birds incapable of flying properly and cats constantly drooling and running in circles [30]. According to the locals, the death of cats, dogs, and pigs was a recurrent event during this period, resembling a similar pattern to the increase of human patients [32].

A second episode of MeHg poisoning happened in 1965, derived from the same acetaldehyde production factory. As a measure to diminish the MeHg content in Minamata Bay, discharges were directed into the Agano River, which caused another outbreak that affected Niigata [5]. This incident is popularly known as the “Niigata Minamata Disease”. Chisso Co. Ltd. eventually ceased functions in May 1969 [7]. As this event was the first case of MeHg intoxication caused by environmental pollution, it took many years to establish the cause of the outbreak [7]. The recognition of mercury’s environmental impacts and acceptance of its global distribution needed decades of expert research [6]. Preservation and analysis of children’s umbilical cords tissue born in the region were crucial to assess MeHg exposure levels and estimate the time-course alterations in MeHg pollution [7]. 

### 2.4. Additional Impactful Events of MeHg Poisoning

A similar incident to the Minamata Disaster occurred in Iraq in the early 1970s [9]. Thousands of cases were admitted to hospitals with severe intoxication, and many passed away [7,36]. The consumption of bread made from wheat seed treated with MeHg-based fungicides was determined as the cause of this epidemic [5]. Several incidents of poisoning have been reported in Pakistan and Guatemala due to the ingestion of flour and wheat seed treated with MeHg compounds [36]. In the same decade, residents of Grassy Narrows, Ontario, Canada showed evidence of neurological alterations which met the criteria for MeHg poisoning, after mercury contamination of the local aquatic system [5].

Local Brazil populations are also exposed to small-scale gold mining activities and mangrove pollution [3,37,38]. The region of Tapajós is one of the largest gold production areas in Brazil, making it a site of chronic exposure to mercury compounds. Human communities living in this environment have been chronically exposed since the 1980s, especially those living in riverside areas. It is believed that working conditions and fish consumption are at the core of MeHg accumulation, raising levels of organic compounds to 70% of total Hg in the blood of mining workers [3,11]. Locals were described with alterations of motor and visual performances, as well as teratogenesis and carcinogenesis, though the latter were difficult to relate to MeHg due to their multifactorial origin [11]. Recently, also in Brazil, two significant environmental disasters occurred in two important ore tailings dams in Minas Gerais (Brumadinho and Mariana) [39]. Thus, the exposed population has become vulnerable to the deleterious low (chronic) level of MeHg, which can lead to low-intensity cognitive impairments, neurodegenerative diseases, and premature aging. This important public health issue should raise the awareness of the Brazilian governmental authorities about the consequences of human exposure to MeHg [39,40].

Other impactful historical events related to MeHg poisoning have been documented around the world, such as the United States, Somalia, Russia, China, England, or Zimbabwe [9,41].

### 2.5. The Minamata Convention

It is unclear how anthropogenic actions will affect the Hg cycle in the years to come. Forestry practices and wildfires may affect watershed Hg processes and MeHg bioaccumulation with a direct impact on terrestrial and aquatic ecosystems. Both oceans and freshwater compartments may be influenced by climate-induced alterations. The tendency for MeHg generation may be increased as a consequence of organic matter remineralization due to increasing seawater temperatures. Regarding the atmospheric reservoir, Hg concentrations are decreasing in Europe and North America whereas the opposite trend happens in Asia [16].

Reports from the United Nations stated that Hg emissions were increasing in developing countries due to coal-burning and gold mining activities [10]. Actions on Hg products and waste could have a beneficial impact in minimizing local exposure to Hg, mainly in developing countries and specific populations such as miners, fishermen, and their families [14]. Regulations and emission control technologies could be valuable to enhance the capabilities of countries to address this problem by implementing safe handling and disposal of Hg containing products [6,14]. Protection of the food-web from exposure to mercury compounds is an important mission for the protection of the population [6]. Other actions that may have a positive impact include the use of clean energy sources that do not rely on coal combustion and switching to non-mercury thermometers and sphygmomanometers in health care [6]. Global greenhouse pollution mitigation measures are believed to contribute to the reduction of anthropogenic Hg emissions [16].

Several international measures have been taken in recent decades to reduce mercury contamination of the environment and ultimately to prevent noxious consequences for ecosystems and human health [14]. The Minamata Convention is a global treaty that signs the commitment by governments of more than 140 nations to diminish Hg emissions and usage in order to reduce harm to both humans and the environment [10,14,42,43]. Nevertheless, it is not clear if the implementation of measures will effectively lead to ecological improvements [14]. Choices made by governments will certainly influence future actions to mitigate Hg emissions and control. However, predictions state that the global Hg cycle will continue to be vulnerable to emissions for decades [16].

## 3. Toxicokinetic Properties of MeHg

Toxicokinetics provides knowledge about how the body affects a toxicant through absorption, distribution, metabolism, and excretion in a time-dependent manner. In general, the toxicity of a xenobiotic depends on the chemical form, time, dose, and pathway of exposure as well as on individual susceptibility to compound hazards [44]. Because of the hazard of MeHg and its long half-life within the human body, the knowledge of its toxicokinetic behavior is important to address risk assessment [45]. From a chemical point of view, MeHg is characterized as a soft acid because the acceptor atom is of low positive charge, large size and has several easily excited outer electrons. This molecule has a high affinity for nucleophilic groups, such as -SeH and -SH groups presented on the structure of several biomolecules [46].

Consequently, the toxicokinetic behavior of MeHg is ultimately conditioned by the chemical profile of this organomercurial compound. The electrophilic nature of MeHg facilitates the reaction with several thiol groups leading to S-mercuriation of several proteins, allowing its transfer between nucleophilic molecules. These interactions modify the protein structure, oxidative state, and biological function [47]. 

Although all mercurial forms have known deleterious effects on human health at high doses, recent evidence indicates an association between chronic low doses of MeHg and cytotoxic effects, mainly in the central nervous system (CNS) [48,49,50]. Pathological examination of patients poisoned with MeHg showed that brain tissue loss is mainly found in occipito-temporal lobes, cortical areas, and cerebellum [51,52]. The damage of MeHg on the occipital lobe, including the primary visual area, or the primary auditory area in the temporal lobe is responsible for alterations in visual field and loss of auditory acuity, common symptoms after a mercuric-related disaster [53,54]. Visual disturbances might be correlated with MeHg binding to outer segments of photoreceptor cells [54]. Regarding the noxious effect on cerebellum, the most susceptible described region to MeHg toxicity, loss of cerebellar glutamatergic granule cells is potentially associated with ataxia, locomotor impairment and occasionally tonic seizures. A recent case report study described several glial reactivity and neuronal loss the occipital lobe and cerebellum in a postmortem toxicological analysis of a 40-year-old man who suffered a fatal intoxication caused by injection of a fluid containing organic mercury [55].

### 3.1. Absorption and Distribution

After consumption of MeHg-contaminated food, the gastrointestinal tract absorbs approximately 95% of ingested MeHg. Therefore, the gastrointestinal tract is considered the main route of MeHg absorption and the primary gateway for MeHg toxicity [56]. Nevertheless, it should be noted that MeHg can also be readily absorbed through the skin and lungs where the extent of absorption is believed to be high (Figure 1A) [57].

Once absorbed into the bloodstream, the majority of MeHg quickly enters the red blood cells (RBC). Although the exact mechanism remains uncertain, one possible explanation could be that MeHg is actively taken up into RBC by multiple transport systems including organic anion transporters [58]. The binding to several cysteine (Cys) residues of α and β chains of hemoglobin (Hb) suggests that Hb could be the major RBC carrier of MeHg [59]. 

A significant part of MeHg is also transported by several plasma proteins. Serum albumin (Alb) is the most abundant plasma protein to which MeHg covalently binds through sulfhydryl groups, forming a reversible MeHg-Alb conjugate [60,61,62]. In fact, as a consequence of its affinity to thiol groups, MeHg conjugates with sulfhydryl-containing molecules like Hb, Alb, L-Cys, and glutathione (GSH) which increases its water solubility [63,64]. This facilitates the circulatory distribution of MeHg conjugates to several organs, mainly to kidneys and brain. Nevertheless, the precise molecular mechanism of distribution through the body remains unknown. However, it is accepted that the formation of thiol complexes results in the transport across cell membranes [64]. For example, several studies indicate that MeHg enters the endothelial cells of the blood-brain barrier (BBB) conjugated with L-Cys via the L-type amino acid transporter 1 (LAT1) system [12,63,64,65,66]. The formed methylmercury-L-Cys conjugate mimics that of the large neutral amino acid L-methionine, which enables MeHg to cross the BBB through the LAT1 system [63,64]. This conjugation is so specific that the complex formed with the isomer D-Cys is not able to cross the BBB [64]. Therefore, this mechanism facilitates the entering of MeHg into the brain [63]. In several organs, a small fraction of MeHg is gradually demethylated to inorganic mercury. In vitro studies have reported a dose-response relationship between MeHg administration and inorganic mercury accumulation [67,68]. Thus, even though MeHg-contaminated food constitutes the source of exposure, it is frequent to find both organic and inorganic mercurial species within the organism.

### 3.2. Metabolism and Excretion

The liver is considered one of the major organomercurial demethylating organs [69,70]. In fact, hepatic HgSe granules were detected in mammals fed with MeHg-contaminated aliments. This suggests that MeHg is possibly demethylated in the liver [68]. Cohort studies from humans exposed to MeHg for a long period have shown a high concentration of inorganic mercury in brain. These results suggest that a slow rate demethylation occurs in brain [71]. Toxicokinetic studies in human subjects suggest a positive correlation between the rate of MeHg demethylation into inorganic Hg and MeHg elimination rate [68]. Regarding excretion pathways, inorganic Hg can be excreted via bile discharges or via urine, while excretion of MeHg occurs predominantly via feces. Inside hepatocytes, MeHg-binding proteins exchange MeHg with GSH, an antioxidant system, resulting in the MeHg-GS complex that undergoes biliary secretion into the intestine. In the distal gut, the majority of MeHg is reabsorbed via enterohepatic circulation through passive transcellular transport. Methylmercury is slowly broken down in the gut to inorganic mercury, thus being excreted through feces [72,73]. Although fecal excretion is considered the canonical MeHg excretory pathway, urinary excretion of MeHg-S conjugates constitutes a complementary detoxification mechanism [74]. The majority of plasmatic MeHg-Alb is transferred to the kidneys through a non-filtrating peritubular mechanism. Once inside the kidneys, MeHg covalently binds to GS and this conjugate is rapidly degraded to Cys-MeHg and excreted through urine. In mice orally administrated with MeHg, MeHg was excreted in the urine predominantly as a Cys conjugate, Cys-MeHg [75]. In the whole body, the half-life elimination is approximately 70–80 days, but it also varies according to doses, time of exposure and individual susceptibility. Therefore, predictive biomarkers associated with individual susceptibility are needed to monitor the progression of MeHg toxicity. Recently, plasma thiol antioxidant capacity was proposed as a future predictive biomarker that could identify and protect the population from MeHg toxicity [76]. 

### 3.3. Influence of Microbiota

Different factors including dietary habits, antibiotic consumption, probiotic intake, genetic variabilities, and individual composition of the gut microbiome influence MeHg toxicokinetic properties [77]. The consumption of fiber-enriched food may be associated with lower levels of MeHg (re)absorption into tissues due to higher detoxification rate [78]. Additionally, the use of antibiotics could be related to MeHg decomposition due to its direct effect on microbiota [79]. Therefore, the gut microbiome seems to contribute to MeHg excretion by unclear mechanisms [77]. It has been suggested that microbial MeHg degradation involves a set of enzymes: mercuric reductase (MerA) and organomercurial lyase (MerB) [80]. However, genetic analysis of human stool samples with low and high levels of MeHg did not reveal the presence of both MerA and MerB, suggesting an alternative demethylating metabolic pathway [77]. In opposition, several toxicological effects of MeHg are expected after prolonged exposure to MeHg, which includes gut dysbiosis, impairment of the barrier function of intestinal epithelium and gut inflammation [72,81].

### 3.4. MeHg Exposure Prevention and Therapeutic Strategies

MeHg presents serious risks by having a long half-life due to the affinity to sulfhydryl-groups present in several structures of the human body [45,60,61,62]. Limitation of the consumption of MeHg-contaminated aliments, especially fish, is the best preventive strategy for avoiding MeHg intoxication and reducing the risk of MeHg-related toxicity. The consumption of mercury-containing fish should be moderated in pregnant women due to an association between in utero exposure and developmental neurotoxicity among populations that frequently consume seafood [82]. According to Minamata Convention, governments are required to address mercury emissions to air and phase out products containing high levels of mercury [14,42]. Whether victims are suffering acute or chronic exposure needs to be determined, and the identification and removal of the mercury exposure source must be conducted promptly [83]. 

Regarding acute exposure to MeHg, current standard clinical strategies for mitigating the toxic effects of MeHg rely almost exclusively on the use of chelators such as meso-2,3-dimercaptosuccinic acid (DMSA) or 2,3-dimercapto-1-propanol (BAL) [84,85]. 

However, therapeutic efficacy is limited, and thus new options are necessary to replace and/or complement the current strategies. Additionally, the use of chelating agents is an unspecific systemic treatment that removes both essential and toxic metals. Therefore, the administration of chelators may cause adverse side effects, depending on the clinical history of patient [86].

Interestingly, data from the last few decades suggest the use of both natural and synthetic antioxidant compounds to reverse the MeHg-induced neurotoxicity, indicating that this class of molecules may be also useful as a pharmacological adjuvant of the current therapies. Moreover, dietary supplementation with Vitamin E and selenium have been shown to have a protective effect when the brain is exposed to MeHg [87,88].

The recent concern regarding the health impact of mercury exposure has been not only directed towards high-intoxicated acute cases but also calling attention to chronic low-grade intoxicated ones. However, the development of treatments for mercury intoxication caused by mercury exposure has been overlooked, especially for long-term exposure cases [7]. New treatment protocols and combination therapies are needed for the appropriate management of methylmercury and mercury poisoning [89].

## 4. Cellular and Molecular Mechanisms Involved in MeHg Neurotoxicity

Understanding how MeHg leads to cytotoxicity is crucial to determine the process that underlies biological perturbations that lead up to the recognized signs of MeHg intoxication [8]. Despite the numerous studies, the mechanisms by which MeHg induces toxicity have not been fully disclosed.

MeHg-induced neurotoxic effects are intimately related with the capacity of this compound to readily disturb and cross BBB (Figure 1B) [90,91,92]. Growing in vitro and in vivo evidence indicates an association between MeHg exposure and BBB dysfunction, with immunoglobulin extravasation, decreased expression of endothelial cell antigen-1, and vascular endothelial growth factor (VEGF) upregulation [93,94]. In fact, VEGF and VEGF receptor-1/-2 expression is enhanced in endothelial cells upon MeHg exposure, which is relevant since VEGF is able to induce vessel hyperpermeability and subsequently vascular leakage [93,94]. Moreover, MeHg leads to delayed maturation of vessels which translates into a defective barrier property [90]. 

The developing nervous system seems to be sensitive to noxious effects of MeHg, but the adult system can also be affected, particularly following long-scale environmental accidents. Organotypic cultures of rat cerebellar cultures exposed to MeHg showed a delay in synaptogenesis, impairment in cell migration, and a disorganized cerebellar architecture [95,96,97]. Since prenatal exposure to MeHg is implicated in childhood cognitive deficit, especially in learning and memory, hippocampal neurogenesis has been pointed out as a likely vulnerable target of MeHg noxious effect. In fact, despite the non-existence of clear evidence in humans, in vivo experimental models reported pathophysiological alterations in hippocampus following both perinatal and post-natal MeHg exposure [98,99,100]. 

At the cellular level, MeHg has been correlated with changes in oxidative stress, increase in excitotoxicity, deoxyribonucleic acid (DNA) damage, alterations in neurogenesis, calcium (Ca^2+^) dyshomeostasis, exacerbation of neuroinflammation, and concomitantly cell death mechanisms (Figure 2) [9,12,90,101,102,103].

MeHg appears to play key roles in the in vivo and in vitro pathological process of MeHg intoxication induced by oxidative stress. Conversely, several studies have reported a partial amelioration of MeHg toxicity following both enzymatic and non-enzymatic antioxidant administration [104,105,106,107,108]. Nevertheless, the molecular mechanisms underlying MeHg-mediated oxidative stress are not yet fully understood. Under physiological conditions, the disruption of redox homeostasis by an excess of reactive oxygen species (ROS) formation or a depletion in antioxidant defenses leads to a cumulative oxidative stress. Antioxidant factors as the reduced glutathione (GSH)/oxidized glutathione (GSSG), thioredoxin reductase (Trx), glutathione peroxidase (GPx), glutathione reductase (GR), and nuclear factor erythroid 2–related factor 2 (Nrf2) are described as pivotal players in maintaining redox balance in the brain, a tissue prone to oxidative stress due to high-energy demand. GSH is a ROS scavenger system that acts as a buffer system and thus limiting the amount of MeHg available for interaction with other biomolecules [109].

Increased intracellular levels of MeHg is associated with both increased and decreased GSH levels. MeHg was shown to interfere with both Nrf2 and the uptake of Cys, key steps involved in GHS synthesis [110]. This suggests that MeHg can increase ROS, which may either inhibit GSH levels or initiate an adaptive response to oxidative stress by increasing GSH levels. Even so, GSH modulation by MeHg seems to rise the susceptibility of certain neuronal cells towards MeHg-induced oxidative stress [9,12]. This may put in motion an escalating oxidative stress which drives MeHg neurotoxic effects. Additionally, higher levels of GSH may constitute an advantage cytoprotective mechanism against MeHg toxicity since these cells may have an enhanced excretory pathway with a consequent higher resistance to MeHg.

### 4.1. Microglia

Microglia are the resident immunocompetent cells of the CNS [111]. These cells have a major role in maintaining homeostasis by responding to environmental alterations [112,113,114]. Under normal conditions, microglial cells survey the surrounding parenchyma by extending and retracting processes [114,115]. After an insult, microglia respond with changes in morphology, proliferation, migration, phagocytic activity, and secretion of a plethora of cytokines and growth factors [113,114,116,117,118]. Microglial cells take part in important neurological processes such as neurogenesis and synaptic pruning [119,120,121]. Many studies have assessed the effects of MeHg exposure in numerous features of microglial cells in many biological systems as models of MeHg intoxication (Appendix A). The results vary according to the experimental protocol and model established to expose these cells to MeHg. Nevertheless, these studies comprise valuable information for the understanding of the role of microglia in MeHg-induced toxicity.

Microglia viability is an addressed key feature regarding MeHg-induced toxicity. In vitro experiments reveal a decrease in microglial viability which is time- and concentration-dependent. Additionally, the biological model of microglia used in the experiments may determine the magnitude of the effect of MeHg in cell viability [122,123,124,125,126,127,128,129]. However, some researchers indicate the absence of microglial death and clustering surrounding apoptotic cells in aggregating brain cell cultures [130]. Nishioku et al. [122] revealed that caspase-3 activity was involved in the mechanism of cell death under MeHg exposure. Oxidative stress is another mechanism that may be involved in microglia viability and cell function perturbations by MeHg, as described in other cell types [12]. In fact, reports indicate that ROS are enhanced in microglial cells as early as 10 min and up until 60 min of exposure to MeHg (1–40 µM) [124,129]. In agreement with these results, ROS increase was also detected in rat glioma cells incubated with MeHg in the same concentration and time range [131]. Additionally, Ni and colleagues observed a decline in the GSH/glutathione disulfide (GSSG) ratio, meaning that GSH is being oxidized and indicating the production of ROS [129]. Nrf2 is also increased after MeHg exposure as a possible cellular mechanism to counteract the upsurge of ROS [116,118]. Interestingly, the KO of Nrf2 further exacerbated the loss of cell viability induced by MeHg [129]. In addition, Garg and colleagues disclosed that the prostaglandin derivate 15-deoxy-delta 12, 14-prostaglandin J2 was able to attenuate the reduction in cell viability due to its antioxidant activity [124]. Another alteration related to MeHg-induced oxidative stress is the increased mitochondrial depolarization and the reduction of aconitase activity and cytoplasmic content in microglia [124]. Interestingly, exposure to a low concentration (2ng/mL) of MeHg for 48 h was able to induce an increase in microglial cells viability, contrarily to what is observed with higher MeHg concentrations [125]. 

Microglial morphology alterations are another important issue in MeHg neurotoxicity studies. Actually, these alterations provide information regarding its activity state since they occur in response to toxic stimuli and/or alterations in the brain parenchyma [132,133]. In the presence of MeHg, microglia display an activated morphology characterized by a round shape with shorter, thicker, and less numerous processes in microglial primary cultures, organotypic cerebral cortex slices, hippocampal sandwich cultures, and animal models [122,123,134,135,136,137]. Sholl analysis and time-lapse recordings show a reduction in morphological complexity and process extension and retraction activities [134]. Amoeboid microglia with increased cell body size and circularity are features of microglia with an activated state following MeHg incubation [122,135,136,137]. It was suggested that MeHg has the ability to increase Rho-associated coiled-coil containing protein kinase (ROCK) signaling in microglia [134,138]. This is consistent with microglia morphological alterations since ROCK is a regulator of the actin cytoskeleton [134]. Regarding microglial activity state, Charleston et al. [103] noticed that the chronic administration of MeHg in adult female monkeys was able to elicit an increase in reactive microglia in the visual cortex. Other brain parenchyma cell types were left unaltered [103]. Moreover, an increase in the mercury content of microglia relative to other cells was also reported using the same experimental protocol. The same authors further showed that mercury detection inside microglia occurred earlier than in other parenchyma cell types [139]. Other studies also indicate that MeHg affects microglial reactivity and number. For example, Monnet-Tschudi and colleagues observed an increase in the number and clustering of microglia in rat aggregating brain cell cultures exposed for 10 days to MeHg [140]. Additionally, the development of clusters was detected in rat primary microglial cells upon exposure to MeHg for 5 days [123]. Moreover, Shinozaki et al. [134] revealed an increase in ionized calcium-binding adapter molecule 1 (Iba1) staining on organotypic cortical slices, which indicates microglial reactivity. The mRNA levels of CD16 and CD32, markers of microglial reactivity were also found to be enhanced [135]. Accordingly, several animal studies indicate that chronic MeHg administration has an effect in the enhancement of microglial cells staining, reactivity, and number in several areas of the CNS, including occipital lobes [141], prelimbic and motor cortex [66], cerebellum [137,142], hippocampus [136], dorsal root ganglions and sensory fibers [143], dorsal root nerve and dorsal column of the spinal cord [138]. However, the increased staining detected in spinal nerves and sensory fibers may be derived from macrophages recruited to the area. Nevertheless, microglial cells account for the most part of the detected signal [143]. 

Function of microglia, including the production and release of cytokines and inducible nitric oxide synthase (iNOS) expression, is affected by MeHg as well. Regarding cytokines, interleukin 6 (IL-6) is one of the most studied in this context. It was demonstrated that MeHg exposure produced an increase in IL-6 both in animal models and microglial cell lines studies [124,131,138]. Conversely, a decline in this cytokine has been detected in microglial primary cultures [123,126]. On the other hand, Shinozaki et al. [92] could not detect alterations in IL-6. These apparent discrepancies may derive from different experimental models and protocols. Nonetheless, MeHg seems to induce changes in this cytokine. Other studies showed an increase in mRNA and protein levels of interleukin-1 beta (IL-1β) [125,138]. Additionally, mRNA and protein levels of the tumor necrosis factor (TNF) seem also to increase in microglia following a MeHg stimulus [125,134,138]. In line with these findings, Iwai-Shimada and colleagues detected an increase in TNF mRNA levels in the cerebrum and cerebellum which, despite not being confirmed, may be of microglial origin [144]. However, MeHg at a low concentration (2 ng/mL) for 48 h exerted a reduction in microglial cells TNF mRNA levels [125]. Another study revealed no alterations in TNF levels [123]. 

Regarding the nitric oxide (NO) pathway, increased levels of iNOS have been described in animal studies, primary microglial cultures, and organotypic cortical slices [134,138]. Whereas one study could not detect alterations in NO levels [123], another study showed that an elevated concentration of MeHg could decrease NO and a lower concentration induced the opposite [125].

Finally, other functional studies were performed. In particular, it was demonstrated that MeHg triggers the phosphorylation of p38 mitogen-activated protein kinase (MAPK) which leads to the release of Adenosine triphosphate (ATP) to the extracellular compartment via vesicular nucleotide transporters [92] and phosphorylation of nuclear factor kappa-light-chain-enhancer of activated B cells (NF-κB) p65 was also documented [138]. Intracellular Ca^2+^ levels were also addressed by Tan and colleagues revealing a similar pattern to NO, with Ca^2+^ increasing under the lower concentration of MeHg and decreasing with the higher concentration of MeHg [125].

It is important to notice that microglia seem to be more sensitive to MeHg, since it responds earlier to lower concentrations of MeHg than other cells. Microglia may have a protective profile under these conditions [123,127,128,129,130,139]. For example, there is strong evidence that MeHg is primarily sensed by microglia communicating and interacting synergistically with astrocytes in order to secrete neuroprotective molecules in an attempt to protect neurons from cell death [123,127,128,129,130,139]. However, higher concentrations of MeHg may tip the balance towards a neurotoxic microglia profile. 

### 4.2. Oligodendrocytes

Little has been published concerning either Hg or MeHg effects on oligodendrocytes (Appendix A). Studies in female monkeys reveal no alterations in the number of oligodendrocytes after MeHg exposure [103]. However, a minority of scattered oligodendrocytes accumulated MeHg intracellularly [139]. A severe and swift decline in the rate of oxygen uptake was detected in purified cultures of oligodendrocytes upon MeHg incubation [145]. Additionally, the activity of 2′,3′-cyclic-nucleotide 3′-phosphodiesterase, an enzyme expressed exclusively by oligodendrocytes, was found significantly reduced in aggregating cell cultures of fetal rat telencephalon upon treatment from days 5 to 15 with 1 µM MeHg. This indicates that MeHg can exert cytotoxic effects in this cell type [140]. Issa and colleagues observed that metal ions (including Hg2+) induce cytotoxic effects in MO3.13 oligodendrocyte cell line [146]. In this case, MeHg and Hg induced a dose-dependent loss of cell viability, with MeHg being more toxic [146]. A couple of recent studies on human brains denoted the accumulation of Hg inside the cerebral cortex and geniculate nuclei oligodendrocytes [147,148]. Mercury in oligodendrocytes may possibly affect axonal conductance and myelin metabolism which in turn have the ability to decelerate nerve impulses [147].

### 4.3. Astrocytes

Astrocytes are the focus of great scientific interest, because they are the largest and most prevalent type of glial cells in CNS and thus have been widely studied regarding MeHg toxicity (Appendix A). 

Astrocytes contribute to the formation of the BBB, support neuronal development and plasticity, and ensure the maintenance of extracellular ionic and chemical balance, including glutamate homeostasis, by tiling the entire CNS [149]. Similarly to microglia, after an insult, astrocytes suffer biochemical and structural alterations and release a cocktail of pro-inflammatory mediators that may exert either neuroprotective or neurotoxic effects depending on the type, duration, and severity of insult [150]. 

MeHg tends to diffuse across astrocytic membranes through the LAT1 system, thus altering astrocyte function [128]. Regarding morphology, immunohistochemistry analysis reported changes in glial fibrillary protein (GFAP), a marker for astrocytic activity, associated with a hypertrophic and swelled morphology with shorter and thicker processes and cell shrinkage, a pro-inflammatory associated phenotype [151]. 

A recent study evaluating MeHg in the inferior colliculus of mice exposed to MeHg described an increase in in the expression of GFAP, indicating astrogliosis activation associated with a neuroprotective response [152]. Equally, subcutaneous injections of MeHg increased the levels of S100 calcium-binding protein B (S100β), a Ca^2+^ binding protein physiologically produced and released by astrocytes, in cerebrospinal fluid, [153]. Since inhibition of astrocytic activation has been shown to accelerate neuronal damage, it is believed that reactive astrogliosis could represent a primary neuroprotective response against MeHg-induced oxidative stress [91]. In co-culture model of astrocytes and neurons challenged with MeHg, neuronal viability was less compromised compared to neuronal monoculture, corroborating the neuroprotective role of astrocytes against MeHg-induced neurotoxicity [154]. Nevertheless, sustained astrocytic activation may provide harmful responses that can lead to chronic neuroinflammation and ultimately compromise the viability of neuronal cells. 

Glutamate is the major excitatory neurotransmitter in the CNS, where it plays key functions in learning, development, and memory. However, it can trigger excitotoxic mechanisms if not properly buffered by astrocytes [155]. In fact, astrocytes are responsible for glutamate clearance from extra synaptic space through excitatory glutamate transporters, mainly glial glutamate transporter-1 (GLT-1) and glutamate/aspartate transporter (GLAST), preventing accumulation in the synaptic cleft and ensuing excitotoxicity [156]. High extracellular levels of glutamate can induce neuronal excitotoxicity due to overactivation of the N-methyl-D-aspartate (NMDA) receptors, leading to increased Ca^2+^ influx into neurons and activation of neuronal apoptosis pathway [157,158]. Besides, glutamate uptake by astrocytes is highly dependent on the electrochemical gradient [156]. Thus, disturbances of glutamate handling may disrupt electrochemical and consequently cytoplasmic osmotic load, a condition known as astrocytic swelling, a hallmark of astrogliosis [159,160]. 

MeHg intoxication alters astrocytic glutamate handling in a time-dependent manner. In astrocytes, MeHg contributes to dysregulation of glutamate uptake and contributes to glutamate efflux via GLAST and GLT-1, leading to an accumulation of extracellular glutamate nearby glial and neuronal cells [161]. Thus, it is likely that MeHg impairs glutamate uptake by covalently binding to SH-groups on Cys residues of glutamate transporters glutathione S-transferase (GST) and GLT-1 [162]. Glutamine is an important precursor for the biosynthesis of glutamate and GSH. Astrocytic-derived glutamine is accumulated by neurons, where it is converted into glutamate, which, in turn, is released to the synaptic cleft and accumulated by astrocytes that convert it into glutamine by glutamine synthetase [163]. The effects of MeHg on glutamine uptake and glutamate cycling has also been studied. In fact, it was found that 10 μM of MeHg reduces the glutamine uptake, altering glutamine/glutamate cycling and contributing to MeHg toxic effect [164,165]. The combination of all effects contributes to the increase of extracellular glutamate synaptic levels which in turn trigger excitotoxicity via N-methyl-D-aspartate (NDMA) receptors [159,166,167,168,169]. 

Oxidative stress, derived from both the generation of ROS and the decrease of antioxidant machinery levels, is another MeHg-mediated mechanism of astrocytic toxicity [170,171]. In primary rat astrocytes, ROS scavenging by antioxidants shows a neuroprotective role in response to acute MeHg exposure via hypoxia-inducible factor 1-alpha (HIF-1α), a transcription factor with critical roles in adaptive and cytoprotective responses. Consistently, the reduction in HIF1-α protein levels in primary rat astrocytes is significantly correlated with a decrease in cell proliferation and an increase in cytotoxicity triggered by MeHg exposure [172,173]. On the other hand, excessive ROS is associated with the activation of phospholipase A2 that produces astrocyte swelling [174]. The astrocytic GSH system has a pivotal role in antioxidation and detoxification of xenobiotics. Astrocytic GPx activity was found to be decreased after MeHg exposure, highlighting the high vulnerability of these cells to MeHg-mediated oxidative stress [175]. Nrf2 is another important factor that is disrupted in astrocytes following MeHg exposure. It was shown that MeHg overactivates Nrf2 and upregulates Nrf2-derived downstream antioxidant genes via phosphoinositide 3-kinase (PI3K)/protein kinase B (Akt) [176]. Therefore, further characterization of MeHg involvement in Nrf2 and PI3K/Akt crosstalk is needed. A suggested mechanism of MeHg-induced astrocytic gliotoxicity implies a decrease in GSH intracellular levels which enhances oxidative stress and may contribute to astrocytic cell death [174,176,177]. S. Robitaille et al. stated that 1 µM of MeHg for 24 h induces a 12-fold increase in GSSG resulting in a disruption in redox homeostasis and aberrant S-glutathionylation of proteins through glutaredoxin-1 overactivation [178]. Moreover, MeHg was reported to selectively impair astrocytic Cys uptake, the rate-limiting GSH substrate, through inhibition of Na^+^-dependent Cys transporters which in turn impairs the activity of GSH system [170]. Thus, GSH pool represents a great target for MeHg toxicity in astrocytes, ultimately altering the redox microenvironment and contributing to astrocytic-mediated neurotoxicity. 

MeHg also affects astrocytes through the increase of intracellular [Ca^2+^], via mitochondria damage and dysregulation of voltage-gated Ca^2+^ channels, as well as Ca^2+^ release from intracellular reservoirs. Once inside the cell, Ca^2+^ is buffered by mitochondria [179]. Due to its role in antioxidative fine-tuned regulation, mitochondria are seemingly a major target of MeHg-mediated astrotoxicity [180]. Hence, several mitochondrial mechanisms including alterations on mitochondrial membrane potential, Ca^2+,^ and metabolic alterations may initiate a cascade of events that culminate in caspase-dependent cell death [165]. Furthermore, oxidative stress activates MAPK signaling pathway through phosphorylation of one of the downstream components, ERK, which can exacerbate cell death-related pathways [181]. Some studies indicate that sustained ERK phosphorylation is upregulated after exposure to MeHg [165]. Alessandra and colleagues showed that when astrocytes are challenged with 10 µM of MeHg for 24 h, the levels of mitochondrial caspase-3 and activated ROCK-1 increased suggesting an involvement of ROCK pathway in mediating apoptosis [182]. For instance, the activation of ROCK-1 by caspase-3 is a step necessary to membrane blebbing during apoptosis [183]. 

Interestingly, recent data propose necrosis as a prevalent mechanism of MeHg-induced cell death in astrocytes via c-Jun N-terminal kinase (JNK)/MAPK signaling pathway, cytoskeletal alterations, and changes in GFAP network [175]. One study suggested that MeHg may induce oxidative stress responsible for caspase-mediated apoptosis and then necrosis ensues [184]. However, further evidence is needed to clarify the exact biochemical pathway by which MeHg triggers apoptosis and late necrosis.

In summary, oxidative stress represents a pivotal event related to MeHg-induced astrotoxicity. Accordingly, redox imbalance due to ROS overproduction and depletion of antioxidant mechanisms, glutamine/glutamate cycling disruption, and Ca^2+^ dyshomeostasis synergistically trigger a set of mechanisms that lead to astrocytic dysfunction and impair astrocytic-neuronal network which ultimately converges into astrocytic death-related pathways.

### 4.4. Neurons

Neuronal cells are extremely sensitive to any neurotoxin that threatens brain homeostasis such as pathogenic organisms or toxic metals, including mercury-derived compounds [185]. 

Although some studies have observed deleterious effects in the peripheral nervous system, the CNS is the main target of MeHg, as evidenced by symptoms following an acute or chronic exposure [143]. For this reason, MeHg-mediated neurotoxicity has been extensively studied using both in vitro and in vivo biological models (Appendix A). Multiple synergic molecular mechanisms contribute to MeHg neurotoxicity including cytoskeletal alterations, proliferation damage, oxidative stress, changes in Ca^2+^ signaling, mitochondrial damage, and glutamate-mediated excitotoxicity [165,186,187,188].

MeHg has been recognized to impact the production of new neurons from a pool of radial glial-like neural stem cells (NSC). NSCs are committed to differentiate into newborn neurons, which can be integrated into preexisting neuronal circuits or into glial cells [189]. Important in vitro and in vivo studies have been performed to investigate the effect of MeHg on the developing brain and their concomitant effects on later cognitive abilities. Some of them reported alterations in several cycle-related checkpoints, including S-phase of the mitogenic cycle, leading to a decrease of NSC proliferation and engagement of senescent state [99,190,191]. Low exposure to MeHg affects DNA synthesis in vitro and in vivo, but also induces caspase-dependent cell death in hippocampal granule cells, reinforcing the idea that hippocampal neurons may be particularly vulnerable to harmful effects of MeHg [99,190,192]. Dentate gyrus and hilus regions of the hippocampus are particularly prone to MeHg toxicity. In rats, the number of immature progenitor cells, immunocharacterized by sex determining region Y-box 2 (Sox2), doublecortin (DCX) and bromodeoxyuridine (BrdU) markers, decrease following subcutaneous injection with MeHg. This reduction could be related with apoptosis-driven mechanisms [99,190,192,193]. Interestingly, in vitro data using primary hippocampal cells treated with 100 nM of MeHg for 24 h revealed a decrease in both immature DCX cells and microtubule-associated protein 2 (MAP-2) cells, as well as an increase in GFAP dependent neurotrophic factors [193,194]. Moreover, animal models injected with MeHg show a reduction in NeuN positive cells, a marker for neuron-committed cells [143,195,196]. These emerging results corroborate the idea that MeHg affects hippocampal neurogenesis, probably by increasing glial differentiated cells.

MeHg has detrimental effects on the developing brain when exposed during gestation, causing toxic effects on synaptic transmission and plasticity in the offspring’s brain. In neurons from neonatal pups whose mothers were gestationally exposed to MeHg, immunofluorescence analysis reported alterations in microtubulogenesis markers, including MAP-2 and β3-tubulin, and an increase in GFAP in offspring’s hippocampus. These combined alterations could lead to architectural changes in cytoskeleton and abnormal neurite outgrowth, as shown by a loss in expression of neurite-specific marker neurofilament triplet H protein (NF-H) [197,198,199,200,201,202]. Furthermore, data using in vitro cell lines suggest that low concentrations of MeHg changed α-tubulin and acetylated-tubulin levels in mouse embryonic fibroblasts, one of the most abundant proteins presented in microtubules cytoskeleton [203].

Besides structural maintenance, microtubules are also involved in axonal transport of secretory vesicles and organelles, migration, proliferation, neuronal growth, and differentiation [204]. Consequently, MeHg exposure may disrupt neuronal viability via several microtubules-involved changes. 

In physiological conditions, mitochondria buffer cytosolic Ca^2+^ levels. MeHg was reported to cause mitochondrial dysfunction via interaction with mitochondrial proteins presented in the respiratory chain or via excessive mitochondrial Ca^2+^ influx through Ca^2+^ channels [205,206]. In human neuronal progenitor cells, MeHg overloads mitochondrial Ca^2+^ leading to an impairment in ATP synthesis [207]. Besides their role as a bioenergetic organelle, mitochondria are also involved in oxidative stress processes and programmed cell death via caspase-dependent mechanisms [208,209]. In fact, in vivo and in vitro experiments show a significant increase in caspase-3 and concomitantly reduced cell viability after MeHg exposure suggesting that neuronal mitochondria may be also especially vulnerable to MeHg toxic effects [175]. Other authors have also found that in vitro cells challenged with low doses of MeHg underwent apoptosis in a mechanism driven by ROS-mediated Akt inactivation and up-regulated endoplasmic reticulum stress [210].

Even though GABAergic, dopaminergic, and cholinergic neurotransmission has been pointed as potential targets of MeHg toxicity, the most available data concerning neurotoxic effect of MeHg implies the glutamate neurotransmitter system [211,212].

As previously stated, one of the hallmark effects caused by MeHg in neurons is glutamate-mediated excitotoxicity [169]. Low doses of MeHg can induce glutamate-mediated neuronal death, as neurons are particularly sensitive to glutamate excitotoxicity [213]. MeHg can increase cytosolic Ca^2+^ through interaction with glutamate receptors such as α-amino-3-hydroxy-5-methyl-4-isoxazolepropionic acid (AMPA) and NMDA, causing an entry of Ca^2+^ through neuronal cell membrane [157,202]. Even acute low concentrations of MeHg increase the frequency of spontaneous excitatory postsynaptic currents, indicating a MeHg-induced shift towards hyperexcitability [214].

Furthermore, MeHg was also reported to block the counteracting GABAergic signaling in neurons, the main inhibitory neurotransmitter pathway [215]. Altogether, MeHg promotes an imbalance in excitation/inhibition electrophysiology driven by a glutamatergic overactivation causing hyperexcitability of neurons. Some authors hypothesize that the release of glutamate might be the main mechanism underlying MeHg-induced neuronal death [216]. However, other related mechanisms may also potentiate and ultimately converge into an excitotoxicity driven by glutamate overactivation.

During the process of MeHg-induced excitotoxicity, a decrease of the intracellular GSH level and an increase of ROS production are generally observed. In rat cerebellar granular neurons exposed to MeHg, brain-derived neurotrophic factor (BDNF) was shown to potentiate neurotoxic effects of MeHg, contributing to a more pronounced depletion of GSH [217]. Paradoxically, BDNF has been recognized as a survival factor against apoptosis of neurons and a stimulator of neuronal necrosis and thus being pivotal for shifting from MeHg-induced apoptosis to necrosis. Lower levels of BDNF were also reported following MeHg exposure [201].

Exposure to MeHg directly stimulates SH-SY5Y neuronal-like cells to produce pro-inflammatory cytokines that exacerbate the neuroinflammatory microenvironment, along with caspase-1, caspase-3, caspase-8, and 8-OHdG levels, possibly involving p38 MAPK-mediated oxidative stress and neuro-hyperactivity [218,219]. 

Santana and colleagues found that adult mice chronically exposed to low doses of MeHg display neurodegeneration in the motor cortex with oxidative misbalance, a decrease of neuronal and astrocytic populations, and motor learning impairment [220]. A recent study suggests that low doses of MeHg during a long period are able to trigger several biological impairments related with neurogenesis, learning, and memory processes in the hippocampus of rats, facilitating neurodegenerative events [195]. Since hippocampal neurogenesis-related structural damage is considered an important risk factor for neurodegenerative and psychiatric diseases, these findings suggest that MeHg could exacerbate or trigger neurodegenerative disorders [39]. 

Neurons exposed to low doses of MeHg undergo cell death by the activation of intracellular pathways that lead to apoptosis. Upon exposure of neurons to MeHg, cytochrome c is released from the intermembrane space of the mitochondria and interacts with cytosolic proteins forming an apoptosome complex leading to activation of caspases 9, with subsequent cleavage of the effector caspase-3. In parallel, MeHg-induced neuronal apoptosis can also be triggered by an increase in extracellular and intracellular TNF in a [Ca^2+^]-dependent mechanism [144,221]. Although apoptosis has been recognized as the predominant cell death mechanism in neurons exposed to low concentrations of MeHg, accumulating evidence suggest that higher concentrations of MeHg could also trigger necrosis-related neuronal death [8,51,198]. Depending on the intensity and the duration of the insult, the exacerbation of oxidative stress can cause both types of death in neurons. Regarding oxidative stress, MeHg has been shown to aggravate ROS production and impair antioxidant defenses, even in neurons, resulting in lipidic peroxidation [107,222], contributing to apoptotic-independent cell death. One study reports that early time exposure to 10 µM of MeHg causes mitochondrial activity impairment, disrupting cell membrane integrity, resulting in necrotic cell death in primary cultures of cerebellar granule neurons (CGN). Later, however, CGN undergoes apoptosis following an 18-h treatment [223]. Nevertheless, further data elucidating the neuronal necrotic death triggered by MeHg are needed. It is important to emphasize that MeHg relies on time and dose to induce noxious outcomes and concomitant neuronal loss, contributing, in that way, to later memory deficits [224].

## 5. Conclusions

MeHg exposure has been extensively studied since the Minamata Disaster. It is now evident from the number of cellular functions and organ systems disturbed by MeHg that the exposure has detrimental impacts on public health. This review article summarizes the current data from cellular, animal, and human studies which are pivotal for the clarification of MeHg-induced neurotoxicity and neuroinflammation. In this regard, oxidative stress, cytokine release, mitochondrial dysfunction, glutamate and Ca^2+^ dyshomeostasis, and ultimately cell death are significant consequences of brain cells exposure to MeHg. Although experimental findings have been extremely important in producing novel evidence on aspects involved in MeHg-induced neurotoxicity, additional research is needed to continue clarifying the precise short-term effects of MeHg on brain cells as well as the long-term consequences for human health. Understanding the molecular mechanisms underlying MeHg toxicity has emerged as a public health concern worldwide. However, few studies have evaluated the effects of MeHg exposure on the CNS at equivalent doses to those found in fish-eating populations.

Importantly, more studies addressing the impact of MeHg in the adult neurogenesis process have now emerged. Altogether, the evaluation of the deleterious effect on mice chronically exposed to MeHg constitutes an emerging approach to better understand the pathophysiology across life and the brain long-term consequences of this toxin.

Public health efforts should be made in order to reduce the occurrence of MeHg poisoning and to ensure the correct protection of exposed populations. There is a critical need for the monitorization and intervention actions in such communities, along with legal establishment of MeHg reference doses and clinical intervention procedures. Additionally, public awareness of MeHg intoxication consequences is crucial to avoid unnecessary exposure.

## Figures and Tables

**Figure 1 ijms-22-03101-f001:**
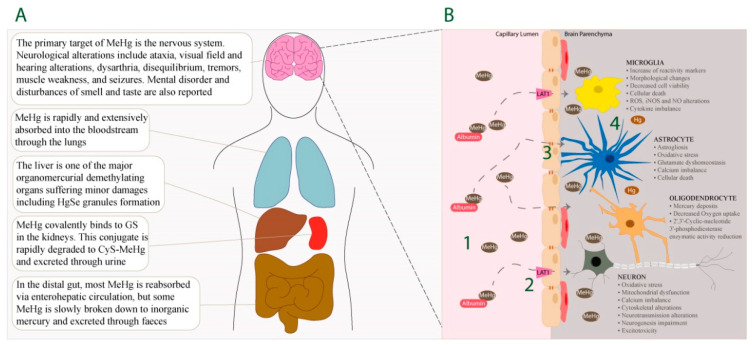
MeHg distribution across the body and main detrimental effects on the brain parenchyma. (**A**) The main organs affected by MeHg include the brain, liver, lungs, kidney, and intestines. (**B**) MeHg circulates in the bloodstream both freely and bound to albumin (1). After conjugating with L-cysteine, MeHg enters the endothelial cells via the LAT1 system (2) to finally cross the BBB (3). Once in the brain parenchyma, MeHg exerts its deleterious effects on neuronal cells and may be demethylated to inorganic Hg (4).

**Figure 2 ijms-22-03101-f002:**
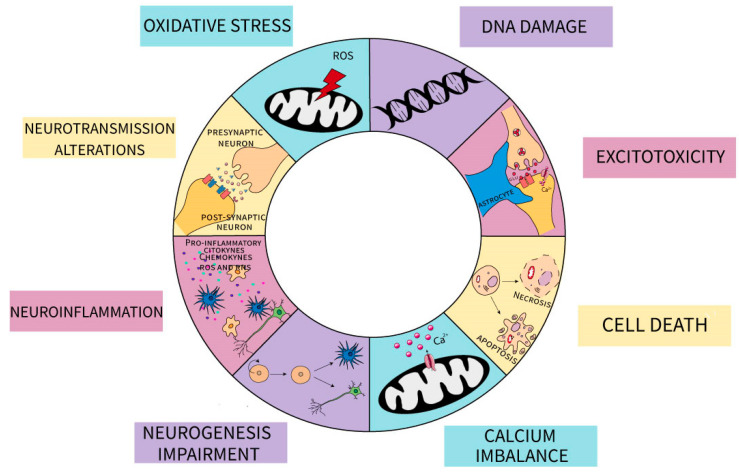
Schematic representation of the impact of MeHg on neuronal and glial cells. The different mechanisms are indicated: oxidative stress mediated by mitochondrial production of reactive oxygen species (ROS) and decrease in antioxidant defenses; DNA damage; excitotoxicity due to changes in both astrocytic and neuronal glutamate metabolism; cell death by apoptotic and necrotic pathways; Calcium imbalance characterize by an increase in [Ca^2+^] cytoplasmatic; neurogenesis impairment both in neuronal and glial-committed lineages; exacerbation in neuroinflammation by an increase in proinflammatory mediators released by both glial and neuronal cells as cytokines, chemokines, ROS and reactive nitrogen species (RNS); changes in synaptic neurotransmission.

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
