# Peer review of "Cellular and Molecular Mechanisms Mediating Methylmercury Neurotoxicity and Neuroinflammation"

_ijms, 2021, doi:10.3390/ijms22063101_

Round 1

Reviewer 1 Report

Dear Authors, 

This is an interesting review regarding the toxicity of methylmercury. It is very well structure, according to the goal of the review. I have some suggestions: 

  1. Figure 1.A., it will be easier to see the ADME process of the methylmercury to see arrows to follow the trail of methylmercury.
  2. Are there any studies explaining strategies to reduce the the exposure of methylmercury or reduce their effects? I think it will be interesting to include a section with these studies. 

Thank you very much. 

Author Response

Dear Reviewer

Many thanks for your constructive comments and suggestions.

After carefully evaluating the pros and cons of modifying Fig.1., we dare to suggest keeping the figure in the present form. We consider that the inclusion of additional arrows will bring visual complexity defocusing from the key messages.

We now introduced a new section (3.4.) highlighting "MeHg exposure prevention and therapeutic strategies".

We take the opportunity to acknowledge the reviewer for these suggestions that contributed to improve the manuscript.

Sincerely,

João O. Malva

Reviewer 2 Report

The review " Cellular and Molecular Mechanisms Mediating Methylmercury Neurotoxicity and Neuroinflammation" by Novo et al., is aimed to provide an overview of the published data concerning the methylmercury neurotoxicity. This manuscript is well organized, first providing important information on distribution, biomagnification, major environmental accidents with serious health consequences, and regulatory issues concerning this environmental pollutant. Information of metylmercury absorption, distribution, metabolism and excretion, as well as its influence on microbiota follows, providing sufficient information for detailed review of the influence this pollutant has on specific populations of cells in central nervous system. Molecular mechanism underlying toxic effects of methylmercury have been summarized, and observed consequences of exposure to low vs. high concentrations of methylmercury have been discussed. This manuscript is well written, pointing to the importance of stringent regulatory polices implementation to confine detrimental effect it has on human health. This review can potentially be interesting to a broad audience.

Author Response

We acknowledge the kind words and positive input provided by the reviewer.

We are confident that the publication of our manuscript will bring value to the scientific community from different research areas.

With my personal best regards,

João O. Malva